# Comparative Long Non-Coding Transcriptome Analysis of Three Contrasting Barley Varieties in Response to Aluminum Stress

**DOI:** 10.3390/ijms25179181

**Published:** 2024-08-23

**Authors:** Xue Feng, Xiaoya Chen, Quan Meng, Ziyan Song, Jianbin Zeng, Xiaoyan He, Feibo Wu, Wujun Ma, Wenxing Liu

**Affiliations:** 1The Characteristic Laboratory of Crop Germplasm Innovation and Application, Provincial Department of Education, College of Agronomy, Qingdao Agricultural University, Qingdao 266109, China18853337206@163.com (Z.S.); hexiaoyan@qau.edu.cn (X.H.); 2Department of Agronomy, College of Agriculture and Biotechnology, Zijingang Campus, Zhejiang University, Hangzhou 310058, China; 3The Key Laboratory of the Plant Development and Environmental Adaptation Biology, Ministry of Education, School of Life Sciences, Shandong University, Qingdao 266237, China

**Keywords:** lncRNA, Al stress, RNA sequencing, Tibetan wild barley, target genes

## Abstract

Aluminum toxicity is a major abiotic stress on acidic soils, leading to restricted root growth and reduced plant yield. Long non-coding RNAs are crucial signaling molecules regulating the expression of downstream genes, particularly under abiotic stress conditions. However, the extent to which lncRNAs participate in the response to aluminum (Al) stress in barley remains largely unknown. Here, we conducted RNA sequencing of root samples under aluminum stress and compared the lncRNA transcriptomes of two Tibetan wild barley genotypes, XZ16 (Al-tolerant) and XZ61 (Al-sensitive), as well as the aluminum-tolerant cultivar Dayton. In total, 268 lncRNAs were identified as aluminum-responsive genes on the basis of their differential expression profiles under aluminum treatment. Through target gene prediction analysis, we identified 938 candidate lncRNA-messenger RNA (mRNA) pairs that function in a cis-acting manner. Subsequently, enrichment analysis showed that the genes targeted by aluminum-responsive lncRNAs were involved in diterpenoid biosynthesis, peroxisome function, and starch/sucrose metabolism. Further analysis of genotype differences in the transcriptome led to the identification of 15 aluminum-responsive lncRNAs specifically altered by aluminum stress in XZ16. The RNA sequencing data were further validated by RT-qPCR. The functional roles of lncRNA-mRNA interactions demonstrated that these lncRNAs are involved in the signal transduction of secondary messengers, and a disease resistance protein, such as RPP13-like protein 4, is probably involved in aluminum tolerance in XZ16. The current findings significantly contribute to our understanding of the regulatory roles of lncRNAs in aluminum tolerance and extend our knowledge of their importance in plant responses to aluminum stress.

## 1. Introduction

Barley (*Hordeum vulgare* L.) ranks as the fourth largest cereal crop worldwide. Despite its sensitivity to aluminum (Al) toxicity, it showcases notable genetic diversity, especially among wild accessions [1]. Long-term domestication has led to the rapid loss of many beneficial alleles in cultivated barley, making it challenging to adapt to diverse environmental stresses. In contrast, Tibetan wild barley, originating from the Qinghai-Tibet Plateau in China, is recognized as one of the ancestral sources of cultivated barley [2], which has been reported to possess extensive genetic diversity and unique mechanisms for survival in harsh environments, including aluminum stress.

Aluminum (Al) is the most abundant metal element found on Earth [3]. Being an amphoteric metal, it can react with strong bases and dissolve in acidic environments. Acidic soils account for 50% of the world’s agricultural lands [3,4]. When the soil pH drops below 5.5, Al and aluminide compounds can form soluble Al^3+^ or Al(OH)^2+^ [5], which are highly toxic to the root apex, leading to severe limitations in water and nutrient absorption [6,7]. Therefore, it is of utmost importance to explore excellent germplasm resources and understand the molecular mechanisms underlying aluminum resistance in order to sustain agricultural productivity.

Non-coding RNAs (ncRNAs), such as microRNAs (miRNAs), long non-coding RNAs (lncRNAs), and circular RNAs (circRNAs), are essential regulators of gene expression [8,9]. LncRNAs, as important regulators of transcription and post-transcriptional processes, have been shown to be related to plant development, hormone signaling, and responses to abiotic stresses [10,11,12]. LncRNAs form a diverse family with lengths exceeding 200 nucleotides and possess limited coding potential [13]. They can be categorized into five types: bidirectional, antisense, intronic, intergenic, and sense lncRNAs [14]. LncRNAs exert their functions by interacting with DNA, RNA, and proteins, thereby influencing epigenetic modifications, transcriptional regulation, and post-transcriptional processes [15]. For instance, in *Arabidopsis*, the antisense lncRNA COOLAIR and intronic lncRNA COLDAIR have been identified to inhibit *FLOWERING LOCUS C* (FLC) gene expression through histone modifications, thereby impacting the vernalization process [16]. In Chinese cabbage, the interaction between bra-miR172a and lncRNA regulates target genes responsible for heat tolerance [17]. In *Medicago truncatula* seedlings, numerous low temperature-responsive lncRNAs have been discovered, and the study investigated the potential regulatory network linking the CBF intergenic lncRNA (MtCIR1) with MtCBF genes under cold stress conditions [13]. Wang et al. [18] reported that phosphate deficiency-induced lncRNAs (PDILs) regulate signaling pathways and transport related to Pi deficiency. Additionally, in *Betula platyphylla*, LncRNA2705.1 and LncRNA11415.1 were found to be involved in Cd tolerance by modulating the expression of target genes such as LDHA and HSP18.1 [19]. Furthermore, a study identified 664 drought-responsive lncRNAs in maize [20], 535 drought-responsive lncRNAs were identified in barley [21] (Qiu et al., 2019), 195 cadmium-responsive lncRNAs were found in barley [22] (Zhou et al., 2023), and 191 drought-responsive lncRNAs were identified in rice [23] (Yang et al., 2022). However, the relationships between lncRNAs, aluminum stress, and their regulatory mechanisms are still not well understood.

In a previous study, we identified two distinct wild barley genotypes: XZ16, which is aluminum-tolerant, and XZ61, which is aluminum-sensitive [24]. This raises the question of whether there are any distinct strategies for aluminum tolerance at the lncRNA transcriptome level in XZ16. Therefore, we conducted lncRNA and RNA sequencing under normal conditions and aluminum stress. This study provides thorough insight into how lncRNAs potentially regulate aluminum tolerance in plants, offering valuable contributions to the development of aluminum-tolerant barley cultivars through breeding efforts.

## 2. Results

### 2.1. LncRNAs in Response to Aluminum Stress

LncRNA sequencing was conducted using the Illumina HiSeq platform by Majorbio Technology Co., Ltd. (Shanghai, China), resulting in the generation of 105,937,050, 107,972,494, and 107,717,356 raw reads for XZ16, XZ61, and Dayton in the control libraries, respectively. Similarly, 106,194,624, 102,756,014, and 109,715,138 raw reads were acquired for the Al stress libraries (Table 1). After removing low-quality reads, a total of 102,743,712, 104,662,554, 104,821,514, 102,680,488, 99,076,632, and 106,974,122 clean reads remained in the six libraries of XZ16-Control, XZ61-Control, Dayton-Control, XZ16-Al, XZ61-Al, and Dayton-Al, respectively. Approximately 70% of the clean reads were successfully mapped to the *Hordeum vulgare* L. genome using the HISAT analysis (Table 1). To identify potential lncRNAs, we applied a filtering approach that excluded mRNAs and transcripts shorter than 200 nt. The left transcripts were further assessed using CPC, CNCI, and pfam (Figure 1A). Analysis of the transcript lengths revealed that the majority of RNA transcripts ranged from 200 to 400 bp or exceeded 1800 bp (Figure 1B). Finally, we discovered 2621 novel lncRNAs and 4488 known lncRNAs (Figure 1C,D).

### 2.2. Target Prediction and Family Classification of LncRNAs

On the basis of different mechanisms of action, lncRNAs can be categorized into cis and trans modes. Here, we predicted a total of 707 lncRNAs that were associated with 780 target genes (Appendix A). Among these target genes, 140 were situated within a 20 kb range downstream of the lncRNA loci, 120 were located within a 10 kb range upstream of the lncRNA loci, and 548 genes had regions that overlapped with the lncRNA loci. Additionally, eight target genes were situated in the 1 kb region upstream of the coding gene, transcribed in the opposite direction to the coding gene. The overlapping forms of lncRNA and target genes can be further classified into sense_exon_overlap, antisense_exon_overlap, antisense_intron_overlap, and sense_intron_overlap.

Furthermore, we conducted lncRNA family classification based on the conserved sequences and secondary structures of the lncRNAs (Appendix A). Out of the 388 classified lncRNAs, 37.1% were assigned to the LSU_rRNA_eukarya family (id: RF02543), 13.7% belonged to the MIR1122 family (id: RF00906), 3.6% were classified under the snoZ102_R77 family (id: RF00359), and 3.4% were assigned to the Plant_SRP family (id: RF01855). The left 418 families, each consisting of an average of 2.6 members, collectively represented 42.2% of the categorized lncRNAs.

### 2.3. Identification of sRNA Precursor in LncRNAs

To identify potential miRNA precursors among the lncRNAs, we compared them to the miRBase database. Our analysis led to the discovery of eight lncRNAs in barley that are likely to serve as miRNA precursors. These lncRNAs include hvu-miR444a, hvu-miR444b, hvu-miR166a, hvu-miR166c, hvu-miR5048a, hvu-miR6205, hvu-miR6190, and hvu-miR159a (Appendix A).

### 2.4. Expression Profile Analysis of LncRNA in Three Barley Genotypes

The expression levels of both lncRNA and mRNA genes were compared between normal and Al stress conditions. The results revealed that both lncRNAs and mRNAs were inhibited under Al stress compared to the control condition (Figure 2), indicating a down-regulation of most Al-responsive genes in response to Al stress.

Furthermore, we identified Al-responsive lncRNAs, with 448, 713, and 393 lncRNA genes showing significant differential expression in XZ16, XZ61, and Dayton, respectively (Figure 3A). This suggests that wild barley exhibits more pronounced physiological and biochemical alterations to accommodate Al toxicity, particularly in the Al-sensitive genotype.

In order to identify Al tolerance-related lncRNAs, we conducted a comparison of Al-responsive lncRNAs in the XZ16, XZ61, and Dayton barley genotypes. In total, 268 lncRNAs were classified as aluminum-responsive lncRNAs. Out of them, 222 lncRNAs showed increased expression in XZ16 but were inhibited or remained unchanged in XZ61 and Dayton under Al treatment (Appendix A), while 46 lncRNAs displayed down-regulation in XZ16 but up-regulation or no change in XZ61 and Dayton (Appendix A). However, the number of target genes was limited. Specifically, in XZ16, 100 target genes associated with 81 Al-responsive lncRNAs were significantly upregulated or downregulated in response to aluminum stress (Appendix A). Within the XZ61 genotype, we identified 122 target genes corresponding to 102 Al-responsive lncRNAs (Appendix A). In the case of the Dayton genotype, 85 target genes linked to 75 Al-responsive lncRNAs were discovered (Appendix A). Additionally, 15 genes were exclusively present in XZ16 (Figure 3B; Table 2). Moreover, we observed four lncRNAs that exhibited decreased expression in XZ16 but remained unaltered or increased in XZ61 and Dayton, with their corresponding target genes inhibited in XZ16 but unaltered or up-regulated in XZ61 and Dayton (Table 2). We identified ten lncRNAs that exhibited upregulation in XZ16, while they either remained unchanged or were downregulated in XZ61 and Dayton, and the target genes displayed a similar expression pattern to these lncRNAs.

### 2.5. Annotation and Enrichment Analysis of LncRNA Target Genes

We utilized six databases (NR, GO, COG, KEGG, SwissProt, and pfam) to annotate both known and novel mRNAs. Out of all mRNAs, 99.22% (28,179 in total) were successfully annotated (Figure 4A). Among them, 10,163 mRNAs were consistently annotated in all six databases (Figure 4B). To further understand the functional significance of differentially expressed genes and lncRNA target genes, we conducted KEGG and GO enrichment analyses. The GO classification annotation revealed that the majority of genes in all three genotypes were associated with a cellular anatomical entity, binding, catalytic activity, and a cellular process (Figure 5). Through KEGG enrichment analysis, we identified two unique lncRNA target genes in XZ16, five in XZ61, and three in Dayton that exhibited significant differences under Al exposure (Figure 6A). In XZ16, the functions of the target genes were primarily related to phenylpropanoid biosynthesis, peroxisome, and starch and sucrose metabolism (Figure 6B). In XZ61, the target genes were associated with RNA phenylpropanoid biosynthesis, peroxisome, necroptosis, and aminoacyl–tRNA biosynthesis (Figure 6C). In Dayton, the target genes were related to phenylpropanoid biosynthesis, peroxisome, phenylalanine metabolism, and ubiquitin-mediated proteolysis (Figure 6D).

To enhance our understanding of the roles played by lncRNA target genes responding to Al stress in XZ16, we linked the differentially expressed genes (DEGs) to protein–protein interaction (PPI) data, leading to the construction of PPI networks. We identified 24 relationships among 23 genes (nodes) in the network (Figure 7, Appendix A). Notably, gene-LOC123447317 played a prominent role in the protein network, suggesting that endochitinase may be involved in aluminum tolerance in XZ16.

### 2.6. Alternative Splicing Analysis

We employed rMATS software v4.3.0 to analyze alternative splicing events, including SE, MXE, A3SS, A5SS, and RI. In total, we identified 12,324, 12,641, and 12,458 alternative splicing events in XZ16, XZ61, and Dayton under Al stress, respectively, compared to the control condition (Figure 8, Appendix A). Notably, alternative splicing events were also observed in differentially expressed genes regulated by differentially expressed lncRNAs. Specifically, we found four alternative splicing events in XZ61 and three in Dayton, involving SE, A3SS, and A5SS. However, no splicing events were observed in XZ16 (Appendix A).

### 2.7. Validation of LncRNA and mRNA Expression Using qRT-PCR

To validate the high-throughput sequencing results, we performed qRT-PCR on selected lncRNAs and mRNAs. Overall, the expression trends in response to Al stress were similar between qRT-PCR and sequencing data. However, specific fold changes observed in qRT-PCR for some lncRNAs and mRNAs did not entirely match the sequencing data. Figure 9 illustrates that the majority of lncRNAs and mRNAs showed consistent results between qRT-PCR and sequencing data.

## 3. Discussion

Long non-coding RNAs (lncRNAs) are increasingly acknowledged for their pivotal roles as regulators in a wide range of biological processes, including the response to abiotic stress in plants. Extensive research has identified numerous abiotic stress-responsive lncRNAs in plants [6,25,26,27,28]. For instance, MtCIR1 has been found to negatively regulate salt stress response in *Medicago truncatula* [29], while lncRNA973 enhances salt stress tolerance in cotton [6]. Qin et al. [30] reported the induction of a lncRNA, DRIR (DROUGHT INDUCED lncRNA), by ABA, drought, and salt stress in Arabidopsis. In acidic soils, aluminum toxicity poses a significant constraint on plant growth [31]. In rice, TCONS_00021861 regulates YUCCA7 by acting as a sponge for miR528-3p. This interaction activates the IAA biosynthetic pathway, thereby enhancing resistance to drought stress [32]. Tang et al. [28] compared drought-responsive genes in two maize, and they found 13 modules were associated with survival rate under drought. Wu et al. [33] compared Al-responsive lncRNAs in two olive (*Olea europaea* L.) genotypes, and they discovered that the targets of the differentially expressed lncRNAs are primarily involved in cell wall modification to enhance aluminum tolerance. Gui et al. [34] identified Al-responsive lncRNAs in *Medicago truncatula*, and found putative target genes were enriched in hormone signal transduction, cell wall modification, and the tricarboxylic acid (TCA) cycle. However, our current understanding remains incomplete without a comprehensive genome-wide identification and characterization of both known and novel lncRNAs linked to aluminum stress in barley, especially in Tibetan wild barley.

The wild barley native to the Qinghai-Tibet Plateau, an ancestral counterpart to cultivated barley, has evolved distinctive mechanisms and harbors extensive genetic diversity. These adaptations enable it to thrive in extreme environments and undergo natural selection [2]. LncRNAs perform various regulatory functions in the modulation of gene expression, including chromatin structure modulation, cis/transcriptional regulation, and post-transcriptional regulation. Previous studies have identified antisense lncRNAs and their impact on sense genes in poplar under nitrogen deficiency [14] and alfalfa under salt stress [35]. In this study, we focused on the Al-tolerant Tibetan wild barley genotype XZ16 to explore Al-related lncRNAs and genes in response to aluminum stress, comparing them with the Al-sensitive Tibetan wild barley and Al-tolerant cv. Dayton. We identified a total of 2621 novel lncRNAs (Figure 1), with 15 differentially expressed lncRNAs likely playing crucial roles in contributing to aluminum tolerance in XZ16 (Table 2). However, the detailed interaction mechanisms and regulatory patterns between these lncRNAs and their putative target genes require further experimental confirmation, as their activation or repression patterns may be more intricate compared to the well-established reverse regulation relationship observed in miRNA-mediated gene regulation.

Long non-coding RNAs (lncRNAs) have been shown to function as precursors for miRNAs, regulating gene expression in response to various stresses. In poplar experiencing nitrogen deficiency, 9 intergenic lncRNAs were identified as precursors for 11 known miRNAs [36]. Similarly, under cadmium stress in *Brassica napus*, four lncRNAs were found to act as precursors for miR824, miR167d, miR156d, and miR156e [37]. Additionally, in *Musa nana* under drought stress, two lncRNAs were revealed as precursors for miR156 and miR166 [38]. In *Gossypium hirsutum* under salt stress, lnc_973 and lnc_253 were identified as precursors for ghr-miR399 and ghr-miR156e [39]. In this study, we predicted 10 lncRNAs as precursors for miRNAs in barley, including hvu-MIR159a, hvu-MIR444a, and hvu-MIR166a (Appendix A). Previous research has indicated that miR159 expression is altered under drought stress in tomatoes [40], and the inactivation of miR166 in maize contributes to plant development and abiotic stress resistance [41]. Additionally, miR444a was found to be up-regulated under aluminum stress in an aluminum-tolerant genotype [42]. The study indicates that lncRNAs play a significant role in miRNA-mediated regulation under aluminum stress, highlighting their importance in plant stress responses.

The function of lncRNAs relies on their interactions with downstream genes. Li et al. [43] demonstrated that CRIR1 (a cold-responsive intergenic lncRNA 1) recruits MeCSP5 (cassava cold shock protein 5) to enhance mRNA translation efficiency, thereby improving cold tolerance in cassava. Sun et al. [44] discovered that the spliceosome MSTRG.85814.11 exerts a positive regulatory effect on its target gene SAUR32, enhancing the rhizosphere response to iron deficiency in apples. In our study, we identified lncRNA-target genes associated with aluminum tolerance, such as disease resistance RPP13-like protein 4, cinnamyl alcohol dehydrogenase, beta-glucosidase, endochitinase, and lectin-domain containing receptor kinase SIT2 (Appendix A). In maize, disease resistance protein ZmRPP13-like 3 (ZmRPP13-LK3) possesses conserved adenylyl cyclase (AC) catalytic center motifs and is significantly induced by 3′-5′-cyclic adenosine monophosphate (cAMP) under heat stress. Engaging with ZmABC2, a presumed cAMP exporter, ZmRPP13-LK3 plays a role in the ABA-mediated response to heat stress [45]. As additional adenylate cyclases (ACs) are recognized in plants, and with progress in quantification techniques, cAMP has gained significance as a pivotal signaling molecule in diverse biological processes, encompassing growth, differentiation, photosynthesis, and stress defense [46,47]. Dai et al. [24] revealed that XZ16 maintains stronger ATPase activity under Al stress. In our study, both RPP13-like protein 4 and cis-acting lncRNA were significantly suppressed in XZ16, but not altered in XZ61 and Dayton (Table 2). This suggests that the highly active ATPase can compete with AC for ATP and HvRPP13-like 4 could contribute to aluminum tolerance in XZ16 through lncRNA-mediated regulation.

Cinnamyl alcohol dehydrogenases (CADs) are enzymes responsible for the synthesis of the main monolignols, namely p-coumaryl-alcohol (H), coniferyl-alcohol (G), and sinapyl-alcohol (S), which serve as essential building blocks of lignin [48]. Lignin is essential for maintaining cell wall extensibility and promoting root elongation in challenging environments [49]. In our study, both the lncRNA rna-XR_006611989.1 and its target gene cinnamyl alcohol dehydrogenase showed simultaneous up-regulation in XZ16, suggesting their involvement in aluminum tolerance (Table 2). However, Wu et al. [33] and Gui et al. [34] reported that several Al-responsive lncRNAs were related to genes participating in the cell wall modification, such as pectinase and xyloglucan endotransglucosylase/hydrolase (XTH), but not CAD. It suggests different species showed different lncRNA-mRNA networks when subjected to Al stress. Plant chitinases, belonging to family 19 of glycosyl hydrolases, are typically endochitinases that randomly cleave chitin, generating various chito-oligosaccharides [50]. These chitinases play essential roles in ethylene synthesis, embryogenesis, and response to environmental stresses. They have also demonstrated potential in response to abiotic stresses such as arsenic and cadmium stress in soybean [50]. Additionally, chitinases have been found effective against heavy metal and salinity stresses [51]. In the present study, the lncRNA MSTRG.28048.1 and its target gene endochitinase A-like protein were up-regulated specifically in XZ16 under aluminum stress, and gene-LOC123447317 played a prominent role in the protein network, indicating the involvement of plant chitinases in aluminum tolerance in barley (Table 2). Although beta-glucosidase and lectin domain-containing receptor kinase SIT2 have not yet been reported to be associated with abiotic stress, further investigations will be conducted to explore their potential roles.

## 4. Materials and Methods

### 4.1. Plant Growth and Treatment

We carried out a hydroponic experiment in a greenhouse at Qingdao Agricultural University, Qingdao, China. Three barley accessions were selected including cultivar Dayton and wild barley (XZ16 and XZ61) [42]. Sterilized seeds were germinated for 7 days and then transplanted into 2 L plastic buckets [42]. The seedlings were cultivated in a one-fifth strength Hoagland’s solution [52] and subjected to continuous ventilation. After five days of growth, the roots underwent a 24 h exposure to either 0 or 50 μM aluminum, with a pH of 4.3. Following this exposure, root samples were collected for RNA extraction. In total, 18 root samples (comprising 3 genotypes, 2 treatment conditions, and 3 replicates) were utilized for RNA sequencing.

### 4.2. High-Throughput Sequencing

After RNA extraction using TRIzol reagent (Invitrogen, Carlsbad, CA, USA), we assessed the quality and quantity of the RNA samples using an Agilent 2100 bioanalyzer (Thermo Fisher Scientific, Waltham, MA, USA). Then, the ribosomal RNA was removed using the Ribo-Zero™ rRNA Removal Kit for purification. Following purification, the RNA was fragmented using a fragmentation buffer. Subsequently, the TruSeq*^®^* Stranded kit (Illumina Inc., San Diego, CA, USA) was utilized to synthesize one strand of cDNA, and the double-stranded cDNA was synthesized using DNA polymerase I and RNaseH. End repair and the addition of “A” bases were performed, followed by the incorporation of Illumina adapters. The second strand of cDNA was digested using the UNG enzyme. The final cDNA library was obtained through amplification and purification of the ligated product. Finally, RNA sequencing was carried out using the Illumina HiSeq platform by Majorbio Technology Co., Ltd. (Shanghai, China).

### 4.3. Mapping and De Novo Assembly

The clean reads obtained from the RNA sequencing were aligned and mapped using the HISAT software v2.1.0 (Hierarchical Indexing for Spliced Alignment of Transcripts) [53]. De novo assembly of the transcripts was conducted using the StringTie software v2.2.3 [54]. To compare the identified transcripts with known mRNA and lncRNA sequences, the gffcompare software v0.12.5 [55] was employed. The classification of lncRNAs was conducted by referencing the NONCODE, GreeNC (Green Non-coding), Ensemble, and NCBI (National Center for Biotechnology Information) databases. To consolidate the assemblies, the cuffmerge software v2.2.1 was utilized. In the preliminary screening process, candidate lncRNAs were selected based on criteria such as a minimum length of 200 base pairs, a minimum of two exons, and an open reading frame (ORF) length of up to 300 base pairs.

### 4.4. Bioinformatics Analysis of Transcriptome Data

To assess the coding potential of the identified transcripts, three software tools, namely CPC [56], CNCI [57], and the Pfam database [58], were employed. The scoring thresholds for the software tools were set as follows: CPC threshold at 0.5 and CNCI threshold at 0.5. Transcripts that could not be found in the Pfam database were classified as lncRNAs. Quantitative analysis of the overall expression levels of lncRNAs and mRNAs was performed using the RSEM software v1.1.17 [59]. Differential gene expression analysis was conducted using the DESeq R package v1.10.1 [60], applying a significance threshold of *p* < 0.05 and a minimum log2^(fold-change)^ of ≥1 to detect genes with significant expression differences.

### 4.5. Target Prediction and Family Analysis of LncRNAs

In order to investigate the functions of the identified lncRNAs, we employed a two-fold approach. Firstly, target genes were predicted in both cis and trans-acting described by Kornienko et al. [61]. For cis-acting lncRNAs, we considered those located within a range of 10 kilobases upstream or 20 kilobases downstream of coding genes. We computed two correlation coefficients, namely Spearman and Pearson, to assess the relationship between the expression profiles of lncRNAs and mRNAs, considering values greater than 0.6 as significant correlations. Additionally, we utilized the INFERNAL software v1.1.4 to analyze the lncRNA families. This analysis involved mapping the lncRNAs to the Rfam database based on conserved sequences and secondary structures, following the approach outlined by Nawrocki et al. [62].

### 4.6. Protein–Protein Interaction Network Analysis of Differentially Expressed LncRNAs Targeted Genes in XZ16

To predict protein interactions, we utilized the STRING online website [63]. Subsequently, interaction networks were constructed using the Cytoscape software v3.8 [64].

### 4.7. sRNA Precursor Prediction and Analysis of Alternative Splicing

To identify potential miRNA precursors, lncRNAs were aligned to the miRBase database according to the method described by Kozomara and Griffiths-Jones [65]. Furthermore, to investigate the presence of alternative splicing (AS) events in the three genotypes under aluminum stress conditions, we employed the rMATS software v4.3.0 (http://rnaseq-mats.sourceforge.net/index.html (accessed on 21 October 2023)). This analysis allowed us to classify the variable splicing events and compare the differences in these events among the different samples.

### 4.8. Annotation and Functional Analysis of LncRNA Target Genes

To gain insights into the potential roles of the target genes regulated by the lncRNAs, we conducted enrichment analysis using the GOseqR package v4.4. Similarly, for the differentially expressed protein-coding genes, GO analysis was performed. To identify the enrichment of lncRNA target genes or differentially expressed genes within KEGG pathways, we utilized KOBAS software v3.0 following the method described by Mao et al. [66]. Subsequently, significantly enriched GO terms and KEGG pathways were established using a corrected *p*-value threshold of <0.05 [37].

### 4.9. Quantitative RT-PCR Validation

We chose 3 lncRNAs and 3 mRNAs for RNA sequencing validation using qRT-PCR with a CFX96 system and SYBR Green Supermix (Bio-Rad, Hercules, USA). At the outset, RNA samples underwent reverse transcription using the Takara PrimeScript™ II 1st strand cDNA synthesis kit. Subsequently, the PCR protocol included an initial denaturation at 95 °C for 30 s, followed by 40 cycles of 95 °C for 3 s and 60 °C for 30 s. The specificity of the PCR amplification was assessed through melting curve analysis. We employed the *Actin* gene as a reference. Experiments were replicated six times (three biological replicates and two technical replicates) and relative quantification was performed using the 2^−ΔΔCt^ method [42]. Primer details are provided in Appendix A, with all primers exhibiting efficiency greater than 90%.

### 4.10. Statistical Analysis

Statistical analysis was carried out using Data Processing System software v18.10 (DPS). ANOVA was employed for significance analysis, followed by Duncan’s Multiple Range Test. Results were considered significant at *p* < 0.05.

## 5. Conclusions

This study represents the first comprehensive investigation of Al-responsive lncRNAs and their target genes on a genome-wide scale, comparing wild barley and cultivated barley with distinct levels of Al tolerance. Through analysis of mRNA libraries from XZ16, XZ61, and Dayton under both control and Al conditions, we successfully identified 2621 novel lncRNAs. Furthermore, by analyzing the transcriptome data, we identified 15 lncRNAs associated with Al tolerance in XZ16 and predicted their target genes, including disease-resistance proteins, endochitinases, and receptor kinases. Our findings provide valuable insights into the functional roles of lncRNAs in response to Al stress, shedding light on their potential contributions to Al tolerance mechanisms.

## Figures and Tables

**Figure 1 ijms-25-09181-f001:**
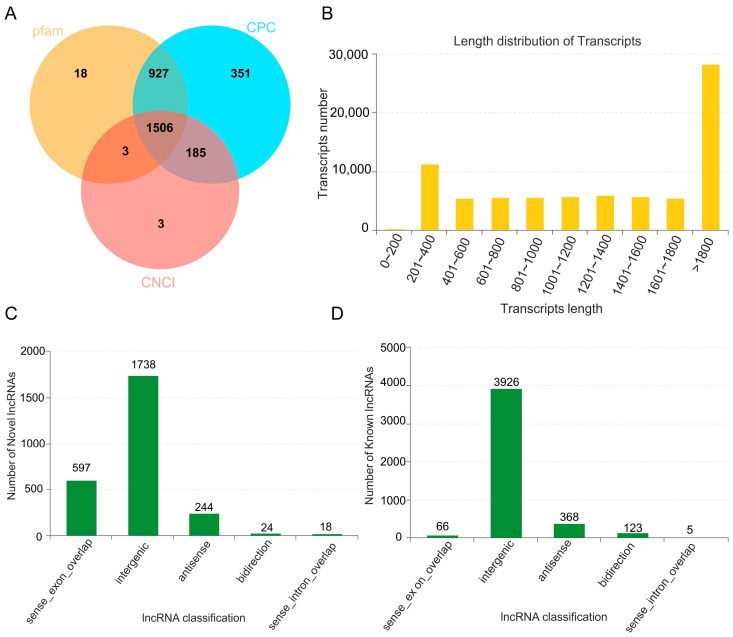
(**A**) The number of novel lncRNA predicted by CPC, CNCI, and pfam. (**B**) Length distribution of lncRNAs and mRNAs. The *X*-axis represents the length of lncRNAs and mRNAs, while the *Y*-axis shows the proportion of corresponding genes. (**C**,**D**) Classification of novel and known lncRNA. sense_exon_overlap: lncRNAs overlapping with exon regions of coding genes; sense_intron_overlap: lncRNAs located within intron regions of coding genes; intergenic: lncRNAs located in intergenic regions between coding genes; antisense: lncRNAs overlapping with the antisense strand of coding genes; bidirection: lncRNAs located within the 1 kb region upstream of coding genes, transcribed in the opposite direction to the coding gene.

**Figure 2 ijms-25-09181-f002:**
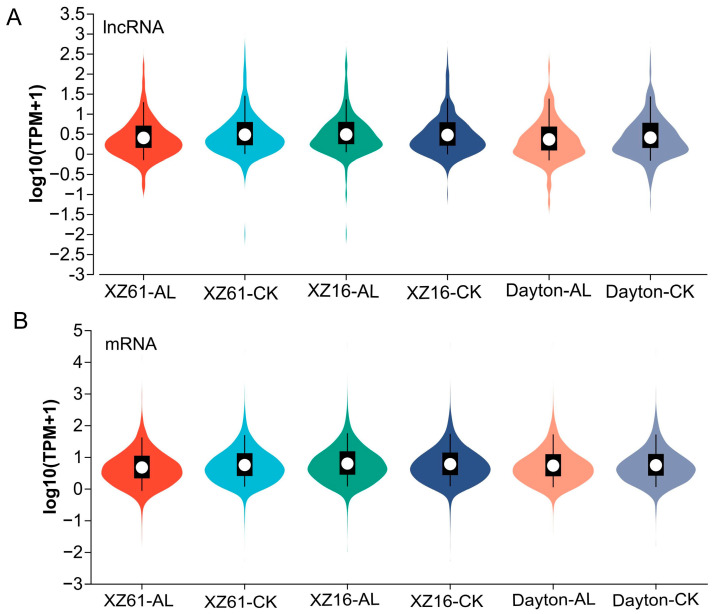
Expression levels of lncRNAs (**A**) and mRNAs (**B**) in six libraries. The *X*-axis indicates RNA density of expression, while the *Y*-axis represents the log10-transformed TPM + 1 values of expression. The enlarged section of the figure highlights the region with the highest concentration of gene expression across the entire dataset. CK presents control, and Al presents aluminum.

**Figure 3 ijms-25-09181-f003:**
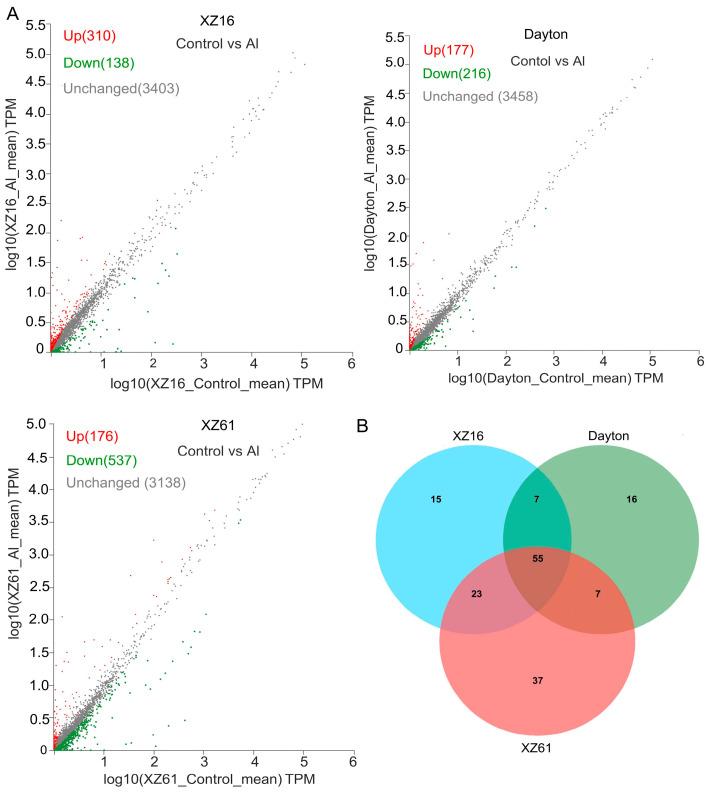
(**A**) Expression differences of lncRNAs under control and Al stress conditions in XZ16, XZ61, and Dayton. The *X*-axis and *Y*-axis represent the log10-transformed TPM values for control and Al treatment, respectively. In the plot, red, green, and gray points denote lncRNAs that are up-regulated, down-regulated, and unchanged under Al treatment compared to control. Fold change (Al vs. control) is represented as log2(N), where log2(N) ≥ 1 indicates up-regulation, |log2(N)| < 1 indicates no change, and log2(N) ≤ −1 indicates down-regulation, with significance at *p* ≤ 0.05. (**B**) Venn diagrams illustrating the number of mRNAs regulated by differentially expressed lncRNAs in three genotypes under Al stress.

**Figure 4 ijms-25-09181-f004:**
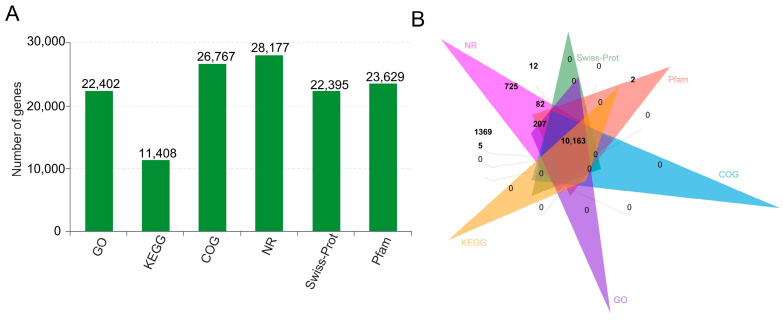
Annotation and functional prediction of all identified mRNAs across six libraries. (**A**) mRNAs were annotated using GO, KEGG, COG, NR, SwissProt, and Pfam databases. (**B**) Venn diagrams illustrate the distribution of mRNAs across the six libraries.

**Figure 5 ijms-25-09181-f005:**
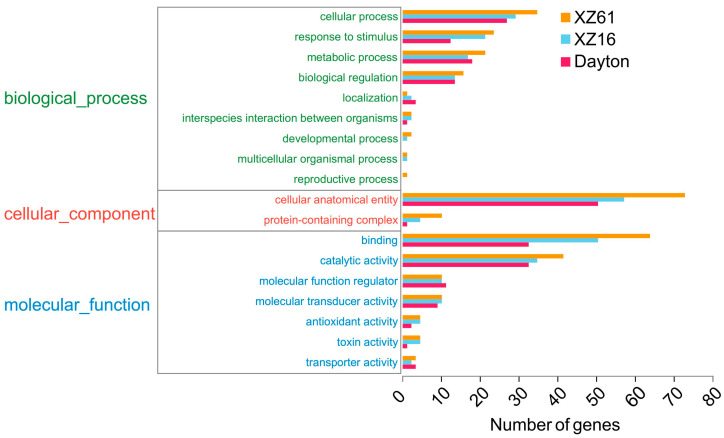
Gene Ontology (GO) analysis was performed on differentially expressed genes. The GO classification of target genes regulated by differentially expressed lncRNAs under Al stress is presented for XZ16 (orange), XZ61 (blue), and Dayton (pink). The *Y*-axis indicates GO terms, while the *X*-axis shows the number of differentially expressed genes (DEGs) associated with each term.

**Figure 6 ijms-25-09181-f006:**
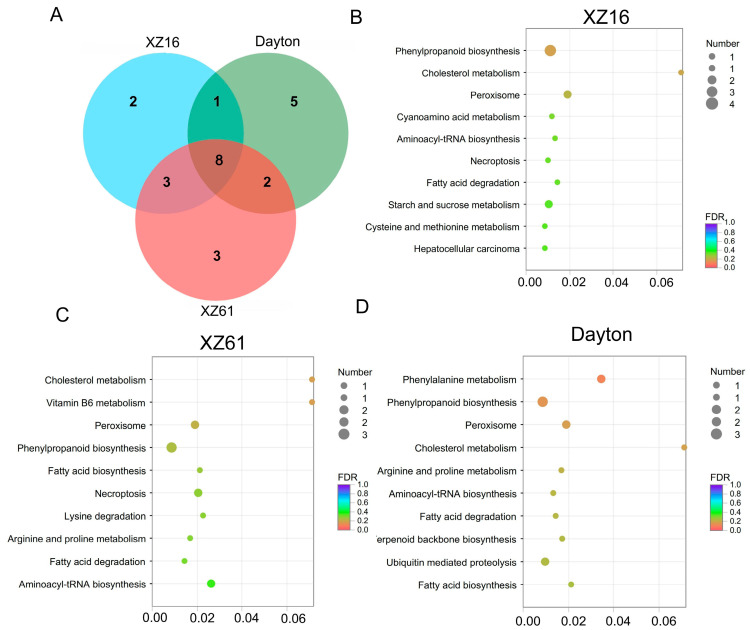
(**A**) Venn diagrams showing the number of target genes regulated by differentially expressed lncRNA in three genotypes under Al stress in KEGG pathway. KEGG enrichment analysis to classify the KEGG pathways associated with target genes regulated by differentially expressed lncRNAs under Al stress in XZ16 (**B**), XZ61 (**C**), and Dayton (**D**). The *Y*-axis represents KEGG pathways, while the *X*-axis indicates the enrichment ratio calculated as the proportion of DEGs among all unigenes enriched in each pathway. The color of each data point indicates the false discovery rate (FDR), and the size of the data point reflects the number of DEGs mapped to the respective pathway.

**Figure 7 ijms-25-09181-f007:**
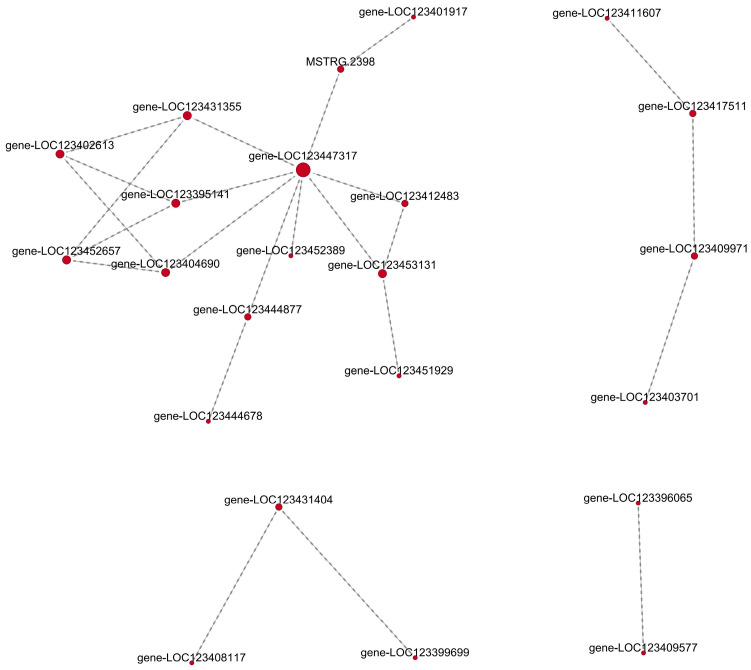
The protein interaction of target genes regulated by differentially expressed lncRNA in XZ16 under Al stress. The gene description is listed in Appendix A.

**Figure 8 ijms-25-09181-f008:**
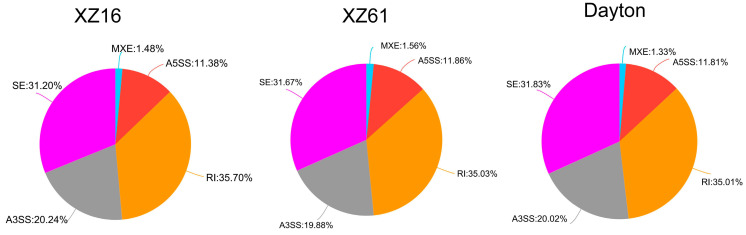
The analysis of alternative splicing events in three barley genotypes. SE: exon jump; A5SS: alternative splicing occurs in the first exon; A3SS: alternative splicing occurs in the last exon; MXE: exon selective hopping; RI: intron retention.

**Figure 9 ijms-25-09181-f009:**
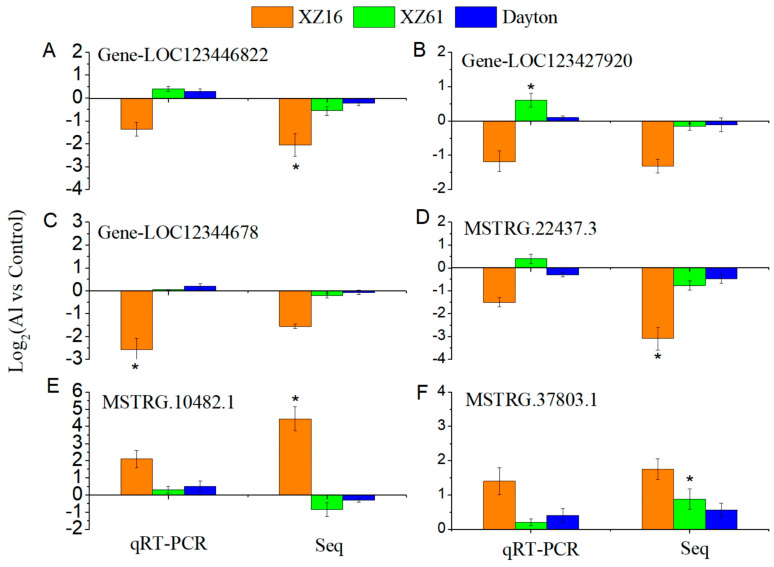
The expression of three mRNAs (**A**–**C**) and three lncRNAs (**D**–**F**) using qRT-PCR in response to Al stress across XZ16 (orange), XZ61 (green), and Dayton (blue). The results are compared with high-throughput sequencing (Seq). Asterisk indicates significant difference.

**Table 1 ijms-25-09181-t001:** Summary of root reads numbers of XZ16, XZ61 and Dayton and their mapped results to barley genome.

Sample	Genotype					
	XZ16		XZ61		Dayton	
	Control	Al	Control	Al	Control	Al
Total Raw reads	105,937,050	106,194,624	107,972,494	102,756,014	107,717,356	109,715,138
Total clean reads	102,743,712	102,680,488	104,662,554	99,076,632	10,4821,514	106,974,122
Total clean reads ratio (%)	98.61	98.61	98.62	98.47	98.58	98.61
Total mapped reads ratio (%)	78.84	79.02	75.0	69.72	76.21	76.55
Uniquely mapping ratio (%)	73.05	74.05	69.11	64.45	69.53	66.35
Novel lncRNA gene	922	925	934	966	890	904
Novel lncRNA isoforms	2177	2186	2219	2279	2166	2174
Konwn lncRNA gene	1876	1873	1889	1922	1922	1923
Known lncRNA isoforms	2497	2515	2516	2541	2564	2543

Control and Al, the plants were treated with basic nutrient solution (BNS) and BNS + 50 μM Al, respectively.

**Table 2 ijms-25-09181-t002:** lncRNAs and their targets simultaneously changed only in XZ16 but not in XZ61 and Dayton.

LncRNA ID	Fold Change (Al vs. Control)			Target mRNA	Fold Change (Al vs. Control)			Annotation
	XZ16	XZ61	Dayton		XZ16	XZ61	Dayton	
MSTRG.22263.1	1.40	0.49	0.00	MSTRG.22262	3.52	2.33	0.00	
MSTRG.32859.1	1.25	−0.97	−0.02	gene-LOC123395388	1.01	0.01	0.36	Uncharacterized LOC123395388
MSTRG.37803.1	1.75	0.88	0.56	gene-LOC123404378	1.32	0.87	0.26	uncharacterized LOC123404378, transcript variant X3
rna-XR_006611989.1	1.87	−0.11	0.66	gene-LOC123404757	1.82	1.23	0.36	probable cinnamyl alcohol dehydrogenase 5
MSTRG.47952.1	1.29	−0.55	1.71	gene-LOC123409748	1.01	0.59	0.39	uncharacterized protein At1g15400-like
MSTRG.47952.1	1.29	−0.55	1.71	gene-LOC123412807	1.03	0.07	0.94	phosphopantothenoylcysteine decarboxylase subunit VHS3-like, transcript variant X1
MSTRG.6428.4	2.42	0.09	−0.02	gene-LOC123421803	1.39	0.52	0.41	uncharacterized LOC123421803
MSTRG.6434.1	3.02	0.92	0.00	gene-LOC123421889	2.81	1.49	1.00	uncharacterized LOC123421889
MSTRG.10482.1	4.43	−0.87	−0.31	gene-LOC123426388	4.22	−0.29	−0.26	uncharacterized LOC123426388
MSTRG.28048.1	2.28	−0.45	−0.33	gene-LOC123447317	2.06	−0.36	−0.31	endochitinase A-like
MSTRG.35329.1	−1.21	−0.33	0.00	gene-LOC123452657	4.39	2.65	0.87	L-type lectin-domain containing receptor kinase SIT2-like
MSTRG.19984.2	−3.33	0.00	1.37	gene-LOC123444678	−1.56	−0.21	−0.07	beta-glucosidase 2-like
MSTRG.22437.3	−3.09	−0.77	−0.48	gene-LOC123446822	−2.05	−0.56	−0.23	disease resistance RPP13-like protein 4, transcript variant X2
MSTRG.22437.3	−3.09	−0.77	−0.48	gene-LOC123446823	−1.98	−0.22	−0.27	disease resistance RPP13-like protein 4
MSTRG.13374.4	−1.07	−0.80	1.22	gene-LOC123427920	−1.32	−0.16	−0.11	putative disease resistance protein RGA4

Control and Al, the plants were treated with basic nutrient solution (BNS) and BNS + 50 μM Al, respectively.

## Data Availability

The datasets generated during the current study are available in the NCBI repository, and the accession number is PRJNA1073472.

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
