# Peer review of "Comparative Long Non-Coding Transcriptome Analysis of Three Contrasting Barley Varieties in Response to Aluminum Stress"

_ijms, 2024, doi:10.3390/ijms25179181_

Round 1

Reviewer 1 Report

Comments and Suggestions for Authors

In my opinion, this study comparing sensitive and tolerant barley cultivars under aluminium stress at the transcriptome level is interesting. Nevertheless, I think that the background should be improved if possible to compare results found in cereals. Comparison between genotypes at results level is not direct in some cases as a lot of information is supplementary. I think you should include the most significant differences in the responses found between the genotypes in the figures (e.g. Fig. 7) and in the main text. I couldn't find the statistics mentioned in point 2.10 in the results. Only two repetitions were performed by condition, this should be justified statistically. Some discrepancies were also reported in lines 323-324, which should be discussed and included as limitations. The quality of the figures should be improved and in some cases the legends should be completed with all the details. The discussion does not focus on the comparison of different genotypes as indicated in the title of this manuscript. Formatting should be checked. The rest of the comments can be found in the attached file.

Comments on the Quality of English Language

In my opinion, only minor English editing is necessary.

Reviewer 2 Report

Comments and Suggestions for Authors

Barly is an important crop throughout the world and the authors put a lot of efforts for gathering and analyzing of the data for this paper. As the sequencing technologies are developing day by day, these studies are strengthening our knowledge and understanding about molecular changes in crops.  However, I would suggest some points, which need to be improved before publication.

The comments are provided for the improvement of each section of the manuscript.

General Comments

Remove the plagiarism of the manuscript, as it is 33% at current stage.

I would suggest to make a graphical abstract as well, to explain key differences in Al tolerance in sensitive and tolerant genotypes.

Abstract

Line 16:  Aluminum can be abbreviated here, and afterwards can be used “Al” in abstract and throughout the manuscript.

Line 21: XZ16 (Al-tolerant) and XZ61 (Al- 21sensitive), please explain that how you the authors determine the tolerance and sensitivity of these genotypes, is there any primarily study conducted? Please explain this thing in Material and methods, and if reference available then provide, or if there is any screeding done, then provide figures and data in supplementary file.

Line 31: second messenger or “secondary messengers” confirm and correct.

Introduction

Lines 69-76: The lines must be moved to the start of the introduction for better clarity.

Material and methods

Line 94: The authors mentioned that “one-fifth strength Hoagland’s solution” was used for the study, don’t you think it is very low strength of headland, is there any study in which this strength of Hoagland solution was used or it is optimized by authors own lab? Please clarify and if there is any reference please cite.

Figures

Figure 1: Please provide the figure 1 in good quality, as it difficult to read in its current form.

Figure 3:  The quality of the figure is not adequate, please increase its quality for better visualization.

Figure 6:  The size of figure A, B, and C is very small, and not readable, please increase the size for better visualization.

Figure 8: Figure 8 text is also very small, and not readable.

Figure 9:  The graph quality and size must be increased so the readers can easily read. Moreover, also provide a, b, c on each graph, and explain in legend.
